# KONG: Kernels for ordered-neighborhood graphs

**Moez Draief**[1]    **Konstantin Kutzkov**[2]    **Kevin Scaman**[1]    **Milan Vojnovic**[2]
[1] Huawei Noah's Ark Lab    [2] London School of Economics, London
`moez.draief@huawei.com, kutzkov@gmail.com` (Corresponding author),
`kevin.scaman@huawei.com, m.vojnovic@lse.ac.uk`

## Abstract

We present novel graph kernels for graphs with node and edge labels that have ordered neighborhoods, i.e. when neighbor nodes follow an order. Graphs with ordered neighborhoods are a natural data representation for evolving graphs where edges are created over time, which induces an order. Combining convolutional subgraph kernels and string kernels, we design new scalable algorithms for generation of explicit graph feature maps using sketching techniques. We obtain precise bounds for the approximation accuracy and computational complexity of the proposed approaches and demonstrate their applicability on real datasets. In particular, our experiments demonstrate that neighborhood ordering results in more informative features. For the special case of general graphs, i.e., graphs without ordered neighborhoods, the new graph kernels yield efficient and simple algorithms for the comparison of label distributions between graphs.

## 1 Introduction

Graphs are ubiquitous representations for structured data and have found numerous applications in machine learning and related fields, ranging from community detection in online social networks [Fortunato, 2010] to protein structure prediction [Rual et al., 2005]. Unsurprisingly, learning from graphs has attracted much attention from the research community. Graphs kernels have become a standard tool for graph classification [Kriege et al., 2017]. Given a large collection of graphs, possibly with node and edge attributes, we are interested in learning a kernel function that best captures the similarity between any two graphs. The graph kernel function can be used to classify graphs using standard kernel methods such as support vector machines.

Graph similarity is a broadly defined concept and therefore many different graph kernels with different properties have been proposed. Previous works have considered graph kernels for different graph classes distinguishing between simple unweighted graphs without node or edge attributes, graphs with discrete node and edge labels, and graphs with more complex attributes such as real-valued vectors and partial labels. For evolving graphs, the ordering of the node neighborhoods can be indicative for the graph class. Concrete examples include graphs that describe user web browsing patterns, evolving networks such as social graphs, product purchases and reviews, ratings in recommendation systems, co-authorship networks, and software API calls used for malware detection.

The order in which edges are created can be informative about the structure of the original data. To the best of our knowledge, existing graph kernels do not consider this aspect. Addressing the gap, we present a novel framework for graph kernels where the edges adjacent to a node follow specific order. The proposed algorithmic framework KONG, referring to Kernels for Ordered-Neighborhood Graphs, accommodates highly efficient algorithms that scale to both massive graphs and large collections of graphs. The key ideas are: (a) representation of each node neighborhood by a string using a tree traversal method, and (b) efficient computation of explicit graph feature maps based on generating $k$-gram frequency vectors of each node's string without explicitly storing the strings. The latter enables to approximate the explicit feature maps of various kernel functions using sketching techniques.

Explicit feature maps correspond to $k$-gram frequency vectors of node strings, and sketching amounts to incrementally computing sketches of these frequency vectors. The proposed algorithms allow for flexibility in the choice of the string kernel and the tree traversal method. In Figure 1 we present a directed labeled subgraph rooted at node $v$. A breadth first-search traversal would result in the string $ABCDGEFHG$ but other traversal approaches might yield more informative strings.

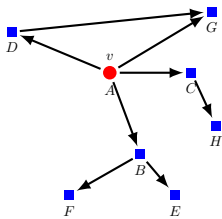

Figure 1: An illustrative example of an order-neighborhood graph: the neighhbor order is the letter alphabetical order.

**Related work**   Many graph kernels with different properties have been proposed in the literature. Most of them work with implicit feature maps and compare pairs of graphs, we refer to [Kriege et al., 2017] for a study on implicit and explicit graph feature maps.

Most related to our work is the Weisfeiler-Lehman kernel [Shervashidze et al., 2011] that iteratively traverses the subtree rooted at each node and collects the corresponding labels into a string. Each string is sorted and the strings are compressed into unique integers which become the new node labels. After $h$ iterations we have a label at each node. The convolutional kernel that compares all pairs of node labels using the Dirac kernel (indicator of an exact match of node labels) is equivalent to the inner product of the label distributions. However, this kernel might suffer from *diagonal dominance* where most nodes have unique labels and a graph is similar to itself but not to other graphs in the dataset. The shortcoming was addressed in [Yanardag and Vishwanathan, 2015]. The kernel between graphs $G$ and $G'$ is computed as $\kappa(G, G') = \Phi(G)^T \mathcal{M} \Phi(G')$ where $\mathcal{M}$ is a pre-computed matrix that measures the similarity between labels. The matrix can become huge and the approach is not applicable to large-scale graphs. While the Weisfeiler-Lehman kernel applies to ordered neighborhoods, for large graphs it is likely to result in many unique strings and comparing them with the Dirac kernel might yield poor results, both in terms of accuracy and scalability.

In a recent work [Manzoor et al., 2016] presented an unsupervised learning algorithm that generates feature vectors from labeled graphs by traversing the neighbor edges in a predefined order. Even if not discussed in the paper, the generated vectors correspond to explicit feature maps for convolutional graph kernels with Dirac base kernels. Our approach provides a highly-scalable algorithmic framework that allows for different base kernels and different tree traversal methods.

The idea of ordered neighborhood was also used in the context of kernels for general graphs. In Martino et al. [2012] it is proposed to decompose the original graph into directed acyclic graphs (DAGs) and then define an order on the DAG nodes which is then used to generate relevant features. The approach was extended to learning the features from graph streams in Martino et al. [2013]. The approaches are not really applicable to our setting as they have a different objective, namely to exploit the structure of general graphs. Also, the DAG generation leads to high computational time.

Another line of research related to our work presents algorithms for learning graph vector representations [Perozzi et al., 2014, Grover and Leskovec, 2016, Niepert et al., 2016]. Given a collection of labeled graphs, the goal is to map the graphs (or their nodes) to a feature space that best represents the graph structure. These approaches are powerful and yield the state-of-the-art results but they involve the optimization of complex objective functions and do not scale to massive graphs.

**Contributions** The contributions of this paper can be summarized as follows:

- To the best of our knowledge, this is the first work to focus and formally define graph kernels for graphs with ordered node neighborhoods. Extending upon string kernels, we present and formally analyse a family of graph kernels that can be applied to different problems. The KONG algorithms are efficient with respect to two parameters, the total number of graphs $N$ and the total number of edges $M$. We propose approaches that compute an explicit feature map for each graph which enables the use of linear SVMs for graph classification, thus avoiding the computation of a kernel matrix of size $O(N^2)$. Leveraging advanced sketching techniques, an approximation of the explicit feature map for a graph with $m$ edges can be computed in time and space $O(m)$ or a total $O(M)$. We also present an extension to learning from graph streams using sublinear space $o(M)$.[1]

- For general labeled graphs without neighbor ordering our approach results in new graph kernels that compare the label distribution of subgraphs using widely used kernels such as the polynomial and cosine kernels. We argue that the approach can be seen as an efficient smoothing algorithm for node labelling kernels such as the Weisfeiler-Lehman kernel. An experimental evaluation on real graphs shows that the proposed kernels are competitive with state-of-the-art kernels, achieving better accuracy on some benchmark datasets and using compact feature maps.

- The presented approach can be viewed as an efficient algorithm for learning compact graph representations. The primary focus of the approach is on learning explicit feature maps for a class of base kernels for the convolutional graph kernel. However, the algorithms learn vector embeddings that can be used by other machine learning algorithms such as logistic regression, decision trees and neural networks as well as unsupervised methods.

**Paper outline** The paper is organised as follows. In Section 2 we discuss previous work, provide motivating examples and introduce general concepts and notation. In Section 3 we first give a general overview of the approach and discuss string generation and string kernels and then present theoretical results. Experimental evaluation is presented in Section 4. We conclude in Section 5.

## 2 Preliminaries

**Notation and problem formulation** The input is a collection $\mathbb{G}$ of tuples $(G_i, y_i)$ where $G_i$ is a graph and $y_i$ is a class. Each graph is defined as $G = (V, E, \ell, \tau)$ where $\ell : V \to \mathcal{L}$ is a labelling function for a discrete set $\mathcal{L}$ and $\tau$ defines ordering of node neighborhoods. We consider only node labels but all presented algorithms naturally apply to edge labels as well. The neighborhood of a node $v \in V$ is $N_v = \{u \in V : (v, u) \in E\}$. The ordering function $\tau_v : N_v \to \Pi(N_v)$ defines a fixed order permutation on $N_v$, where $\Pi(N_v)$ denotes the set of all permutations of the elements of $N_v$. Note that the order is local, i.e., two nodes can have different orderings for same neighborhood sets.

**Kernels, feature maps and linear support vector machines** A function $\kappa : \mathcal{X} \times \mathcal{X} \to \mathbb{R}$ is a valid kernel if $\kappa(x, y) = \kappa(y, x)$ for $x, y \in \mathcal{X}$ and the kernel matrix $K \in \mathbb{R}^{m \times m}$ defined by $K(i, j) = \kappa(x_i, x_j)$ for any $x_1, \ldots, x_m \in \mathcal{X}$ is positive semidefinite. If the function $\kappa(x, y)$ can be represented as $\phi(x)^T \phi(y)$ for an explicit feature map $\phi : \mathcal{X} \to \mathcal{Y}$ where $\mathcal{Y}$ is an inner product feature space, then $\kappa$ is a valid kernel. Also, a linear combination of kernels is a kernel. Thus, if the base kernel is valid, then the convolutional kernel is also valid.

We will consider base kernels where $\mathcal{X} = \mathbb{R}^n$ and $\phi : \mathbb{R}^n \to \mathbb{R}^D$. Note that $D$ can be very large or even infinite. The celebrated kernel trick circumvents this limitation by computing the kernel function for all support vectors. But this means that for training one needs to explicitly compute a kernel matrix of size $N^2$ for $N$ input examples. Also, in large-scale applications, the number of support vectors often grows linearly and at prediction time one needs to evaluate the kernel function for $O(N)$ support vectors. In contrast, linear support vector machines [Joachims, 2006], where the kernel is the vector inner product, run in linear time of the number of examples and prediction needs $O(D)$ time. An active area of research has been the design of scalable algorithms that compute low-dimensional approximation of the explicit feature map $z : \mathbb{R}^D \to \mathbb{R}^d$ such that $d \ll D$ and $\kappa(x, y) \approx z(\phi(x))^T z(\phi(y))$ [Rahimi and Recht, 2007, Le et al., 2013, Pham and Pagh, 2013].

**Convolutional graph kernels** Most known graph kernels are instances of the family of convolutional kernels [Haussler, 1999]. In their simplified form, the convolutional kernels work by decomposing a given graph $G$ into a set of (possibly overlapping) substructures $\Gamma(G)$. For example, $\Gamma(G)$ can be the set of 1-hop subtrees rooted at each node. The kernel between two graphs $G$ and $H$ is defined as $K(G, H) = \sum_{g \in \Gamma(G), h \in \Gamma(H)} \kappa(g, h)$ where $\kappa(g, h)$ is a base kernel comparing the parts $g$ and $h$. For example, $\kappa$ can be the inner product kernel comparing the label distribution of the two subtrees. Known graph kernels differ mainly in the way the graph is decomposed. Notable examples include the random walk kernel [Gärtner et al., 2003], the shortest path kernel [Borgwardt and Kriegel, 2005], the graphlet kernel [Shervashidze et al., 2009] and the Weisfeiler-Lehman kernel [Shervashidze et al., 2011]. The base kernel is usually the Dirac kernel comparing the parts $g$ and $h$ for equality.

Building upon efficient sketching algorithms, we will compute explicit graph feature maps. More precisely, let $\phi_\kappa$ be the explicit feature map of the base kernel $\kappa$. An explicit feature map $\Phi_\kappa$ is

defined such that for any two graphs $G$ and $H$:

$$K(G, H) = \sum_{g \in \Gamma(G), h \in \Gamma(H)} \kappa(g, h) = \sum_{g \in \Gamma(G), h \in \Gamma(H)} \phi_\kappa(g)^T \phi_\kappa(h) = \Phi_\kappa(G)^T \Phi_\kappa(H).$$

When clear from the context, we will omit $\kappa$ and write $\phi(g)$ and $\Phi(G)$ for the explicit maps of the substructure $g$ and the graph $G$.

**String kernels**   The strings generated from subtree traversal will be compared using string kernels. Let $\Sigma^*$ be the set of all strings that can be generated from the alphabet $\Sigma$, and let $\Sigma_k^* \subset \Sigma^*$ be the set of strings with exactly $k$ characters. Let $t \sqsubseteq s$ denote that the string $t$ is a substring of $s$, i.e., a nonempty sequence of consecutive characters from $s$. The spectrum string kernel compares the distribution of $k$-grams between strings $s_1$ and $s_2$:

$$\kappa_k(s_1, s_2) = \sum_{t \in \Sigma_k^*} \#_t(s_1) \#_t(s_2)$$

where $\#_t(s) = |\{x : x \sqsubseteq s \text{ and } x = t\}|$, i.e., the number of occurrences of $t$ in $s$ [Leslie et al., 2002]. The explicit feature map for the spectrum kernel is thus the frequency vector $\phi(s) \in \mathbb{N}^{|\Sigma_k^*|}$ such that $\phi_i(s) = \#_t(s)$ where $t$ is the $i$-th $k$-gram in the explicit enumeration of all $k$-grams.

We will consider extensions of the spectrum kernel with the polynomial kernel for $p \in \mathbb{N}$: for a constant $c \geq 0$,

$$poly(s_1, s_2) = (\phi(s_1)^T \phi(s_2) + c)^p.$$

This accommodates cosine kernel $\cos(s_1, s_2)$ when feature vectors are normalized as $\phi(s)/\|\phi(s)\|$.

**Count-Sketch and Tensor-Sketch**   Sketching is an algorithmic tool for the summarization of massive datasets such that key properties of the data are preserved. In order to achieve scalability, we will summarize the $k$-gram frequency vector distributions. In particular, we will use Count-Sketch [Charikar et al., 2004] that for vectors $u, v \in \mathbb{R}^d$ computes sketches $z(u), z(v) \in \mathbb{R}^b$ such that $z(u)^T z(v) \approx u^T v$ and $b < d$ controls the approximation quality. A key property is that Count-Sketch is a linear projection of the data and this will allow us to incrementally generate strings and sketch their $k$-gram distribution. For the polynomial kernel $poly(x, y) = (x^T y + c)^p$ and $x, y \in \mathbb{R}^d$, the explicit feature map of $x$ and $y$ is their $p$-level tensor product, i.e., the $d^p$-dimensional vector formed by taking the product of all subsets of $p$ coordinates of $x$ or $y$. Hence, computing the explicit feature map and then sketching it using Count-Sketch requires $O(d^p)$ time. Instead, using Tensor-Sketch [Pham and Pagh, 2013], we compute a sketch of size $b$ for a $p$-level tensor product in time $O(p(d + b \log b))$.

## 3   Main results

In this section we first describe the proposed algorithm, discuss in detail its components, and then present theoretical approximation guarantees for using sketches to approximate graph kernels.

**Algorithm**   The proposed algorithm is based on the following key ideas: (a) representation of each node $v$'s neighborhood by a string $S_v$ using a tree traversal method, and (b) approximating the $k$-gram frequency vector of string $S_v$ using sketching in a way that does not require storing the string $S_v$. Given a graph $G$, for each node $v$ we traverse the subtree rooted at $v$ using the neighbor ordering $\tau$ and generate a string. The subtrees represent the graph decomposition of the convolutional kernel. The algorithm allows for flexibility in choosing different alternatives for the subtree traversal. The generated strings are compared by a string kernel. This string kernel is evaluated by computing an explicit feature map for the string at each node. Scalability is achieved by approximating explicit feature maps using sketching techniques so that the kernel can be approximated within a prescribed approximation error. The sum of the node explicit feature maps is the explicit feature map of the graph $G$. The algorithm is outlined in Algorithm 1.

**Tree traversal and string generation**   There are different options for string construction from each node neighborhood. We present a general class of subgraph traversal algorithms that iteratively collect the node strings from the respective neighborhood.

**Definition 1** *Let $S_v^h$ denote the string collected at node $v$ after $h$ iterations. A subgraph traversal algorithm is called a* composite string generation traversal (CSGT) *if $S_v^h$ is a concatenation of a subset of the strings $s_v^0, \ldots, s_v^h$. Each $s_v^i$ is computed in the $i$-th iteration and is the concatenation of the strings $s_u^{i-1}$ for $u \in N_v$, in the order given by $\tau_v$.*

**Algorithm 1:** EXPLICITGRAPHFEATUREMAP.

**Input:** Graph $G = (V, E, \ell, \tau)$, depth $h$, labeling $\ell : V \to \mathcal{L}$, base kernel $\kappa$
**for** $v \in V$ **do**
     Traverse the subgraph $T_v$ rooted at $v$ up to depth $h$
     Collect the node labels $\ell(u) : u \in T_v$ in the order specified by $\tau_v$ into a string $S_v$
     Sketch the explicit feature map $\phi_\kappa(S_v)$ for the base string kernel $\kappa$ (without storing $S_v$)
$\Phi_\kappa(G) \leftarrow \sum_{v \in V} \phi_\kappa(S_v)$
**return** $\Phi_\kappa(G)$

The above definition essentially says that we can iteratively compute the strings collected at a node $v$ from strings collected at $v$ and $v$'s neighbors in previous iterations, similarly to the dynamic programming paradigm. As we formally show later, this implies that we will be able to collect all node strings $S_v^h$ by traversing $O(m)$ edges in each iteration and this is the basis for designing efficient algorithm for computing the explicit feature maps.

Next we present two examples of CSGT algorithms. The first one is the standard iterative breadth-first search algorithm that for each node $v$ collects in $h+1$ lists the labels of all nodes within exactly $i$ hops, for $0 \leq i \leq h$. The strings $s_v^i$ collect the labels of nodes within exactly $i$ hops from $v$. After $h$ iterations, we concatenate the resulting strings, see Algorithm 2. In the toy example in Figure 1, the string at the node with label $A$ is generated as $S_v^2 = s_v^0 s_v^1 s_v^2$ resulting in $A|BCDG|EF|H|G$ ($s_v^0 = A$, $s_v^1 = BCDG$ and $s_v^2 = EFGH$).

**Algorithm 2:** BREADTH-FIRST SEARCH

**Input:** Graph $G = (V, E, \ell, \tau)$, depth $h$, labeling $\ell : V \to S$
**for** $v \in V$ **do**
     $s_v^0 = \ell(v)$
**for** $i = 1$ *to* $h$ **do**
     **for** $v \in V$ **do**
        $s_v^i = \$$     //$\$$ is the empty string
        **for** $u \in \tau_v(N_v)$ **do**
           $s_v^i \leftarrow s_v^i.\text{append}(s_u^{i-1})$
**for** $v \in V$ **do**
     $S_v^h = \$$
     **for** $i = 0$ *to* $h$ **do**
        $S_v^h \leftarrow S_v^h.\text{append}(s_v^i)$

**Algorithm 3:** WEISFEILER-LEHMAN

**Input:** Graph $G = (V, E, \ell, \tau)$, depth $h$, labeling $\ell : V \to S$
**for** $v \in V$ **do**
     **for** $i = 1$ *to* $h$ **do**
        $s_v^i \leftarrow \ell(v)$
**for** $i = 1$ *to* $h$ **do**
     **for** $v \in V$ **do**
        **for** $u \in \tau_v(N_v)$ **do**
           $s_v^i \leftarrow s_v^i.\text{append}(s_u^{i-1})$
**for** $v \in V$ **do**
     $S_v^h \leftarrow s_v^h$

Another approach, similar to the WL labeing algorithm [Shervashidze et al., 2011], is to concatenate the neighbor labels in the order given by $\tau_v$ for each node $v$ into a new string. In the $i$-th iteration we set $\ell(v) = s_v^i$, i.e., $s_v^i$ becomes $v$'s new label. We follow the CSGT pattern by setting $S_v^h = s_v^h$, as evident from Algorithm 3. In our toy example, we have $s_v^0 = A$ and $s_v^1 = ABCD$ and $s_v^2 = ABEFCHDGG$ generated from the neighbor strings $s_u^1, u \in N_v$: $BEF$, $CH$, $DG$ and $G$.

*String kernels and WL kernel smoothing*: After collecting the strings at each node we have to compare them. An obvious choice would be the Dirac kernel which compares two strings for equality. This would yield poor results for graphs of larger degree where most collected strings will be unique, i.e., the diagonal dominance problem where most graphs are similar only to themselves. Instead, we consider extensions of the spectrum kernel [Leslie et al., 2002] comparing the $k$-gram distributions between strings, as discussed in Section 2.

Setting $k = 1$ is equivalent to collecting the node labels disregarding the neighbor order and comparing the label distribution between all node pairs. In particular, consider the following smoothing algorithm for the WL kernel. In the first iteration we generate node strings from neighbor labels and relabel all nodes such that each string becomes a new label. Then, in the next iteration we again generate strings at each node but instead of comparing them for equality with the Dirac kernel, we compare them with the polynomial or cosine kernels. $\cos(s_1, s_2)^p$ decreases faster with $p$ for dissimilar strings, thus $p$ can be seen as a smoothing parameter.

**Sketching of $k$-gram frequency vectors** The explicit feature maps for the polynomial kernel for $p > 1$ can be of very high dimensions. A solution is to first collect the strings $S_v^h$ at each node, then incrementally generate $k$-grams and feed them into a sketching algorithm that computes compact representation for the explicit feature maps of polynomial kernel. However, for massive graphs with high average degree, or for a large node label alphabet size, we may end up with prohibitively long unique strings at each node. Using the key property of the incremental string generation approach

and a sketching algorithm, which is a linear projection of the original data onto a lower-dimensional space, we will show how to sketch the $k$-gram distribution vectors without explicitly generating the strings $S_v^h$. More concretely, we will replace the line $s_v^i \leftarrow s_v^i$.append($s_u^{i-1}$) in Algorithms 2 and 3 with a sketching algorithm that will maintain the $k$-gram distribution of each $s_v^i$ as well as $s_v^i$'s $(k-1)$-prefix and $(k-1)$-suffix. In this way we will only keep track of newly generated $k$-grams and add up the sketches of the $k$-gram distribution of the $s_v^i$ strings computed in previous iterations.

Before we present the main result, we show two lemmas that state properties of the incremental string generation approach. Observing that in each iteration we concatenate at most $m$ strings, we obtain the following bound on the number of generated $k$-grams.

**Lemma 1** *The total number of newly created $k$-grams during an iteration of CSGT is $O(mk)$.*

The next lemma shows that in order to compute the $k$-gram distribution vector we do not need to explicitly store each intermediate string $s_v^i$ but only keep track of the substrings that will contribute to new $k$-grams and $s_v^i$'s $k$-gram distribution. This allows us to design efficient algorithms by maintaining sketches for $k$-gram distribution of the $s_v^i$ strings.

**Lemma 2** *The $k$-gram distribution vector of the strings $s_v^i$ at each node $v$ can be updated after an iteration of CSGT from the distribution vectors of the strings $s_v^{i-1}$ and explicitly storing substrings of total length $O(mk)$.*

The above lemma is the basis for the sketching solution we present next. It will allow us to sketch the $k$-gram distribution vectors at each node and only keep track of the prefixes and suffixes of the strings $s_u^i$.

The following theorem is our main result that accommodates both polynomial and cosine kernels. We define $\cos_k^h(u,v)^p$ to be the cosine similarity to the power $p$ between the $k$-gram distribution vectors collected at nodes $u$ and $v$ after $h$ iterations.

**Theorem 1** *Let $G_1, \ldots, G_M$ be a collection of $M$ graphs, each having at most $m$ edges and $n$ nodes. Let $K$ be either polynomial or cosine kernel with parameter $p$ and $\hat{K}$ its approximation obtained by using size-$b$ sketches of explicit feature maps. Consider an arbitrary pair of graphs $G_i$ and $G_j$. Let $T_{<\alpha}$ denote the number of node pairs $v_i \in G_i$, $v_j \in G_j$ such that $\cos_k^h(v_i, v_j)^p < \alpha$ and $R$ be an upper bound on the norm of the $k$-gram distribution vector at each node.*

*Then, we can choose a sketch size $b = O(\frac{\log M + \log n}{\alpha^2 \varepsilon^2} \log \frac{1}{\delta})$ such that $\hat{K}(G_i, G_j)$ has an additive error of at most $\varepsilon(K(G_i, G_j) + R^{2p}\alpha T_{<\alpha})$ with probability at least $1 - \delta$, for $\varepsilon, \delta \in (0,1)$.*

*A graph sketch can be computed in time $O(mkph + npb \log b)$ and space $O(nb)$.*

Note that for the cosine kernel it holds $R = 1$. Assuming that $p$, $k$ and $h$ are small constants, the running time per graph is linear and the space complexity is sublinear in the number of edges. The approximation error bounds are for the general worst case and can be better for skewed distributions, this follows directly from the properties of the original Count-Sketch algorithm [Charikar et al., 2004].

**Graph streams** We can extend the above algorithms to work in the *semi-streaming graph model* [Feigenbaum et al., 2005] where we can afford $O(n \, \text{polylog}(n))$ space. Essentially, we can store a compact sketch per each node but we cannot afford to store all edges. We sketch the $k$-gram distribution vectors at each node $v$ in $h$ passes. In the $i$-pass, we sketch the distribution of $s_v^i$ from the sketches $s_u^{i-1}$ for $u \in N_v$ and the newly computed $k$-grams. We obtain following result:

**Theorem 2** *Let $E$ be a stream of labeled edges arriving in arbitrary order, each edge $e_i$ belonging to one of $M$ graphs over $N$ different nodes. We can compute a sketch of each graph $G_i$ in $h$ passes over the edges by storing a sketch of size $b$ per node using $O(Nb)$ space in time $O(|E|hkp + b \log b)$.*

The above result implies that we can sketch real-time graph streams in a single pass over the data, i.e. $h = 1$. In particular, for constants $k$ and $p$ we can compute explicit feature maps of dimension $b$ for the convolutional kernel for real-time streams for the polynomial and cosine kernels for 1-hop neighborhood and parameter $p$ in time $O(|E| + Nb \log b)$ using $O(Nb)$ space.

# 4 Experiments

In this section we present our evaluation of the classification accuracy and computation speed of our algorithm and comparison with other kernel-based algorithms using a set of real-world graph datasets. We first present evaluation for general graphs without ordering of node neighborhoods, which demonstrate that our algorithm achieves comparable and in some cases better classification accuracy than the state of the art kernel-based approaches. We then present evaluation for graphs with ordered neighborhoods that demonstrates that accounting for neighborhood ordering can lead to more accurate classification as well as the scalability of our algorithm.

All algorithms were implemented in Python 3 and experiments performed on a Windows 10 laptop with an Intel i7 2.9 GHz CPU and 16 GB main memory. For the TensorSketch implementation, we used random numbers from the Marsaglia Random Number CDROM [mar]. We used Python's scikit-learn implementation [Pedregosa et al., 2011] of the LIBLINEAR algorithm for linear support vector classification [Fan et al., 2008].

For comparison with other kernel-based methods, we implemented the explicit map versions of the Weisfelier-Lehman kernel (WL) [Shervashidze et al., 2011], the shortest path kernel (SP) [Borgwardt and Kriegel, 2005] and the $k$-walk kernel (KW) [Kriege et al., 2014].

**General graphs** We evaluated the algorithms on widely-used benchmark datasets from various domains [Kersting et al., 2016]. MUTAG [Debnath et al., 1991], ENZYMES [Schomburg et al., 2004], PTC [Helma et al., 2001], Proteins [Borgwardt et al., 2005] and NCI1 [Wale and Karypis, 2006] represent molecular structures, and MSRC [Neumann et al., 2016] represents semantic image processing graphs. Similar to previous works [Niepert et al., 2016, Yanardag and Vishwanathan, 2015], we choose the optimal number of hops $h = 2$ for the WL kernel and $k \in \{5, 6\}$ for the $k$-walk kernel. We performed 10-fold cross-validation using 9 folds for training and 1 fold for testing. The optimal regularization parameter $C$ for each dataset was selected from $\{0.1, 1, 10, 100\}$. We ran the algorithms on 30 random permutations of the neighbor node lists and report the average accuracy and the average standard deviation. We set the parameter subtree depth parameter $h$ to 2 and used the original graph labels, and in the second setting we obtained new labels using one iteration of WL. If the explicit feature maps for the cosine and polynomial kernel have dimensionality more than 5,000, we sketched the maps using TensorSketch with sketch size of 5,000.

The results are presented in Table 1. In brackets we give the parameters for which we obtain the optimal value: the kernel, cosine or polynomial with or without relabeling and the power $p \in \{1, 2, 3, 4\}$ of the polynomial (e.g. poly-rlb-1 denotes polynomial kernel with relabeling and $p = 1$). We see that among the four algorithms, KONG achieves the best or second best results. We would like to note that the methods are likely to admit further improvements by learning data-specific string generation algorithms but such considerations are beyond the scope of the paper.

| Dataset | KW | SP | WL | KONG |
|---------|-----|-----|-----|------|
| Mutag | $83.7 \pm 1.2$ | $84.7 \pm 1.3$ | $84.9 \pm 2.1$ | $87.8 \pm 0.7$ (poly-rlb-1) |
| Enzymes | $34.8 \pm 0.7$ | $39.6 \pm 0.8$ | $52.9 \pm 1.1$ | $50.1 \pm 1.1$ (cosine-rlb-2) |
| PTC | $57.7 \pm 1.1$ | $59.1 \pm 1.3$ | $62.4 \pm 1.2$ | $63.7 \pm 0.8$ (cosine-2) |
| Proteins | $70.9 \pm 0.4$ | $72.7 \pm 0.5$ | $71.4 \pm 0.7$ | $73.0 \pm 0.6$ (cosine-rlb-1) |
| NCI1 | $74.1 \pm 0.3$ | $73.3 \pm 0.3$ | $81.4 \pm 0.3$ | $76.4 \pm 0.3$ (cosine-rlb-1) |
| MSRC | $92.9 \pm 0.8$ | $91.2 \pm 0.9$ | $91.0 \pm 0.7$ | $95.2 \pm 1.3$ (poly-1) |

Table 1: Classification accuracies for general labeled graphs (the 1-gram case).

**Graphs with ordered neigborhoods** We performed experiments on three datasets of graphs with ordered neighborhoods (defined by creation time of edges). The first dataset was presented in [Manzoor et al., 2016] and consists of 600 web browsing graphs from six different classes over 89.77M edges and 5.04M nodes. We generated the second graph dataset from the popular MovieLens dataset [mov] as follows. We created a bipartite graph with nodes corresponding to users and movies and edges connecting a user to a movie if the user has rated the movie. The users are labeled into four categories according to age and movies are labeled with a genre, for a total of 19 genres. We

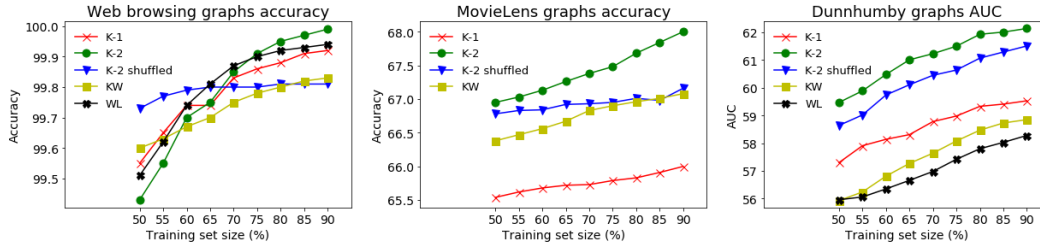
Figure 2: Comparison of classification accuracy for graphs with ordered neighborhoods.

considered only movies with a single genre. For each user we created a subgraph from its 2-hop neighborhood and set its class to be the user's gender. We generated 1,700 graphs for each gender. The total number of edges is about 99.09M for 14.3M nodes. The third graph dataset was created from the Dunnhumby's retailer dataset [dun]. Similarly to the MovieLens dataset we created a bipartite graph for customer and products where edges represent purchases. Users are labeled in four categories according to their affluence, and products belong to one of nine categories. Transactions are ordered by timestamps and products in the same transaction are ordered in alphabetical order. The total number of graphs is 1,565, over 257K edges and 244K nodes. There are 7 classes corresponding to the user's life stage. The classes have unbalanced distribution, and we optimized the classifier to distinguish between a class with frequency 0.0945% and all other classes. The optimal $C$-value for SVM optimization was selected from $10^i$ for $i \in \{-1, 0, 1, 2, 3, 4\}$.

**Results** The average classification accuracies over 1,000 runs of different methods for different training-test size splits are shown in Figure 2. We exclude the SP kernel from the graph because either the running time was infeasible or the results were much worse compared to the other methods. For all datasets, for the $k$-walk kernel we obtained best results for $k = 1$, corresponding to collecting the labels of the endpoints of edges. We set $h = 1$ for both WL and KONG. We obtained best results for the cosine kernel with $p = 1$. The methods compared are those for 2 grams with ordered neighborhoods and shuffled neighborhoods, thus removing the information about order of edges. We also compare with using only 1 grams. Overall, we observe that accounting for the information about the order of neighborhoods can improve classification accuracy for a significant margin. We provide further results in Table 2 for training set sizes 80% showing also dimension of the explicit feature map $D$, computation time (the first value is the time to compute the explicit feature maps and the second value is the SVM training time), and accuracies and AUC metrics. We observe that the WL kernel can generate very long strings, i.e., explicit feature maps of large dimension, which not only lead to the diagonal dominance problem but also result in large computation time; our method controls this by using $k$-grams.

|  | Web browsing | | | MovieLens | | | Dunnhumby | | |
|---|---|---|---|---|---|---|---|---|---|
| **Method** | **D** | **Time** | **Accuracy AUC** | **D** | **Time** | **Accuracy AUC** | **D** | **Time** | **Accuracy AUC** |
| SP | – | > 24 hrs | – – | – | > 24 hrs | – – | 228 | 144" 74" | 90.61 50.1 |
| KW | 82 | 665" 116" | 99.80 99.94 | 136 | 120" 420" | 66.98 73.65 | 56 | 0.7" 134" | 90.57 58.47 |
| WL | 20,359 | 48" 576" | 99.92 99.99 | > 2M | 492" – | – – | 2,491 | 22" 230" | 90.52 57.80 |
| K-1 | 34 | 206" 79" | 99.88 99.97 | 21 | 509" 197" | 65.83 71.00 | 13 | 42" 25" | 90.53 59.33 |
| K-2 shuffled | 264 | 220" 255" | 99.81 99.94 | 326 | 592" 497" | 67.01 73.31 | 85 | 48" 131" | 90.57 61.07 |
| K-2 | 203 | 217" 249" | 99.95 99.99 | 326 | 589" 613" | 67.68 73.20 | 82 | 46" 133" | 90.56 61.94 |

Table 2: Comparison of the accuracy and speed of different methods for graphs with ordered neighborhoods; we use the notation K-$k$ to denote KONG using $k$ grams; time shows explicit map computation time and SVM classification time.

# 5 Conclusions

We presented an efficient algorithmic framework KONG for learning graph kernels for graphs with ordered neighborhoods. We demonstrated the applicability of the approach and obtained performance benefits for graph classification tasks over other kernel-based approaches.

There are several directions for future research. An interesting research question is to explore how much graph classification can be improved by using domain specific neighbor orderings. Another direction is to obtain efficient algorithms that can generate explicit graph feature maps but compare the node strings with more complex string base kernels, such as mismatch or string alignment kernels.

**Acknowledgements.** The work has been supported by a research collaboration grant funded by Huawei Technologies.

## Footnotes

[1]Software implementation and data are available at `https://github.com/kutzkov/KONG`.

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
