[Supplementary Material]

# KONG: Kernels for ordered-neighborhood graphs (Appendix)

**Moez Draief**[1]     **Konstantin Kutzkov**[2]     **Kevin Scaman**[1]     **Milan Vojnovic**[2]
[1] Huawei Noah's Ark Lab    [2] London School of Economics, London
moez.draief@huawei.com, kutzkov@gmail.com (Corresponding author),
kevin.scaman@huawei.com, m.vojnovic@lse.ac.uk

## 1   Additional background

**Count-Sketch and Tensor-Sketch**   We overview sketching concepts that are used in our algorithm.

Count-Sketch [Charikar et al., 2004] was originally proposed as a space-efficient algorithm for frequent items mining in data streams. Let the input be a stream $\mathcal{S}$ of item-weight pairs $(i, w_i)$ and $W = \sum_{(i, w_i) \in \mathcal{S}} w_i$ be the total weight of all items in the stream. Each item can appear arbitrarily many times in the stream. The goal is to detect each item $i^*$ such that $W_{i^*} := \sum_{(i, w_i) \in \mathcal{S}: i = i^*} w_i \geq \gamma W$ for some $\gamma \in (0, 1)$, the so called heavy hitters.

The Count-Sketch algorithm works by distributing the set of items $\mathcal{I}$ to $b$ different bins by a random hash function $h : \mathcal{I} \to [b]$. For each bin $j$, we keep a counter $cnt_j$ which is updated as $cnt_j \leftarrow cnt_j + \text{sign}(i) w_i$ where $\text{sign} : \mathcal{I} \to \{-1, 1\}$ is a random function. After processing the stream, it holds $\mathbb{E}[\text{sign}(i^*) cnt_{h(i^*)}] = W_{i^*}$, i.e., we have an unbiased estimator for $i^*$'s weight. The variance of the estimator decreases with the sketch size $b$. The intuition is that for large enough sketch size $b$, the heavy hitters are likely to be isolated in different bins and the contributions from other items will cancel out due to the sign function.

Count-Sketch can be also used to approximate the inner product of two vectors $x$ and $y$ as $\sum_{j \in [m]} cnt_j^x cnt_j^y$ by sketching each vector separately; $cnt_j^x$ and $cnt_j^y$ denote the corresponding counts for bin $j$. The additive error is bounded by $\|x\| \|y\| / b$. Thus, we can approximate the polynomial kernel by incrementally computing the $p$-level tensor product and then sketching it. However, for $d$-dimensional vectors, this results in $O(d^p)$ running time per input vector, which can be prohibitively expensive already for small values of parameter $p$.

Tensor-Sketch, introduced by [Pagh, 2013],[Pham and Pagh, 2013], is a sketch for $p$-level tensor products, designed by building upon work on approximating matrix multiplication. Using suitably defined hash functions and an application of polynomial multiplication by the fast Fourier transform, it allows to compute the Count-Sketch with $b$ bins of the $p$-level tensor product in time $O(p(d + b \log b))$. Hence, this sketching algorithm has computation time that scales linearly with the input vector dimension. We use this sketch in our algorithm to provide a scalable solution.

## 2   Proofs

**Lemma 1** *The total number of newly created $k$-grams during an iteration of CSGT is $O(mk)$.*

**Proof:** By definition, the string $s_v^i$ generated by CSGT at a node $v$ in the $i$-th iteration is the concatenation of strings generated at its neighbor nodes in the $i - 1$-th iteration. Therefore, new $k$-grams can only be created when concatenating two strings. For a node $v$ there are $|N_v| - 1$ string concatenations, each of them resulting in at most $k - 1$ $k$-grams. Thus, the total number of newly

created $k$-grams is at most

$$\sum_{v \in V} (k-1)(|N_v| - 1) = O(mk).$$

$\square$

The next lemma shows that in order to compute the $k$-gram frequency vector we don't need to explicitly store each intermediate string $s_v^i$ but only keep track of the substrings that will contribute to new $k$-grams and $s_v^i$'s $k$-gram frequency vector.

**Lemma 2** *The $k$-gram frequency vector of the strings $s_v^i$ at each node $v$ can be updated after an iteration of CSGT from the frequency vectors of the strings $s_v^{i-1}$ and explicitly storing substrings of total length $O(mk)$.*

**Proof:** We need to explicitly store only the substrings that will contribute to new $k$-grams. Consider a string $s_v^i$. We need to concatenate the $|N_v|$ strings $s_u^{i-1}$. Since we need to store the $k-1$-prefix and $k-1$-suffix of each $s_u^{i-1}$, for all $u \in V$, it follows that the total length of the stored state is at most

$$\sum_{v \in V} 2(k-1)|N_v| = O(mk).$$

$\square$

**Theorem 1** *Let $G_1, \ldots, G_M$ be a collection of $M$ graphs, each having at most $m$ edges and $n$ nodes. Let $K$ be either polynomial or cosine kernel with parameter $p$ and $\hat{K}$ its approximation obtained by using size-$b$ sketches of explicit feature maps. Consider an arbitrary pair of graphs $G_i$ and $G_j$. Let $T_{<\alpha}$ denote the number of node pairs $v_i \in G_i$, $v_j \in G_j$ such that $\cos_k^h(v_i, v_j)^p < \alpha$ and $R$ be an upper bound on the norm of the $k$-gram distribution vector at each node.*

*Then, we can choose a sketch size $b = O(\frac{\log M + \log n}{\alpha^2 \varepsilon^2} \log \frac{1}{\delta})$ such that $\hat{K}(G_i, G_j)$ has an additive error of at most $\varepsilon(K(G_i, G_j) + R^{2p}\alpha T_{<\alpha})$ with probability at least $1 - \delta$, for $\varepsilon, \delta \in (0, 1)$.*

*A graph sketch can be computed in time $O(mkph + npb \log b)$ and space $O(nb)$.*

**Proof:** First we note that for non-homogeneous polynomial kernel $(x^T y + c)^p$ and $c > 0$, we can add an extra dimension with value $\sqrt{c}$ to the $k$-gram frequency vector of each string. Therefore in the following w.l.o.g. we assume $c = 0$.

We first show how to incrementally maintain a sketch of the $k$-gram frequency vector of each $s_v^i$. In the first iteration, we generate a string $s_v^1$ at each node $v$ from the labels $\ell(u)$ for $u \in N_v$. We then generate the $k$-grams and feed them into $sketch_v$ and keep the $k-1$-prefix and $k-1$-suffix of each $s_v^1$. By Lemma 2, we can compute the $k$-gram frequency vector of $s_v^2$ from the prefixes and suffixes of $s_u^1$, for $u \in N_v$ and the $k$-gram frequency vector of $s_u^1$.

A key property of Count-Sketch is that it is a linear transformation, i.e. it holds $CS(x + y) = CS(x) + CS(y)$ for $x, y \in \mathbb{R}^d$. Thus, we have that

$$CS(s_v^i) = \sum_{u \in N_v} CS(s_u^{i-1}) + CS(K(\tau_v(N_v)))$$

where $K(\tau_v(N_v))$ denotes the newly created $k$-grams from the concatenation of the strings $s_u^{i-1}$.

By Lemmas 1 and 2, we can thus compute a single Count-Sketch that summarizes the $k$-gram frequency vector of $S_v^h$ for all $v \in V$ in time $O(mkh)$ and space $O(nb)$ for sketch size $b$.

For the cosine kernel with parameter $p = 1$, we extend the above to summarizing the normalized $k$-gram frequency vectors as follows. As discussed, Count-Sketch maintains $b$ bins. After processing all $k$-grams of a string $s$, it holds $cnt_j = \sum_{t \in \Sigma_k^* : h(t) = j} \#_t(s)$, where $\#_t(s)$ is the number of occurrences of string $t$ in $s$. Instead, we want to sketch the values $\#_t(s)/w$ where $w$ is the 2-norm of the $k$-gram frequency vector of $S_v^i$. From each Count-sketch $CS(s_v^i)$ we can compute also an $(1 \pm \varepsilon)$-approximation $\tilde{w}$ of the norm of $k$-gram frequency vector [Charikar et al., 2004]. Using that $(1 + \epsilon)/(1 - \epsilon) \geq 1 + 2\epsilon$ and $(1 - \epsilon)/(1 + \epsilon) \geq 1 - 2\epsilon$ for $\varepsilon \leq 1/2$, we can scale $\varepsilon$ in order to obtain the desired approximation guarantee.

Now consider the polynomial and cosine kernels with parameter $p > 1$. Let $TS(x)$ denote the Tensor-Sketch of vector $x$. By the main result from [Pham and Pagh, 2013], for a sketch size $b = 1/(\alpha^2 \varepsilon^2)$, $TS(x)TS(y)$ is an approximation of $(x^T y)^p$ such that the variance of the additive error is $((x^T y)^{2p} + (\|x\|\|y\|)^{2p})/t$. For $\alpha \leq \cos(x, y)^p$ we thus have

$$\frac{(x^T y)^{2p} + (\|x\|\|y\|)^{2p}}{t} \leq \varepsilon^2 \alpha^2 (x^T y)^{2p} + \varepsilon^2 \alpha^2 (\|x\|\|y\|)^{2p} \leq 2\varepsilon^2 (x^T y)^{2p}.$$

A standard application of Chebyshev's inequality yields an $(1 \pm \varepsilon)$-multiplicative approximation of $(x^T y)^p$ with probability larger than 1/2. On the other hand, for $\alpha > \cos(x, y)^p$ we bound the additive error to $2\alpha\varepsilon(\|x\|\|y\|)^p = O(\alpha\varepsilon R^{2p})$. The bounds hold with probability $\delta$ by taking the median of $\log(1/\delta)$ independent estimators, and by the union bound $\delta$ can be scaled to $\delta/(Mn^2)$ such that the bounds hold for all node pairs for all graphs. The same reasoning applies to the cosine kernel where the norm of the vectors is bounded by 1.

The Tensor-Sketch algorithm keeps $p$ Count-sketches per node and we need to feed the $k$-gram distribution vector at each node into each sketch. After $h$ iterations, the $p$ sketches at each node are converted to a single sketch using the Fast Fourier transform in time $O(pb \log b)$. This shows the claimed time and space bounds. $\qquad\square$

## 3 Experiments

In Table 1 we provide additional experimental results on more datasets that confirm that KONG is competitive with the state-of-the-art kernels.

| Dataset | KW | SP | WL | KONG |
|---------|-----|-----|-----|------|
| Mutag | $83.7 \pm 1.2$ | $84.7 \pm 1.3$ | $84.9 \pm 2.1$ | $87.8 \pm 0.7$ (poly-rlb-1) |
| Enzymes | $34.8 \pm 0.7$ | $39.6 \pm 0.8$ | $52.9 \pm 1.1$ | $50.1 \pm 1.1$ (cosine-rlb-2) |
| PTC | $57.7 \pm 1.1$ | $59.1 \pm 1.3$ | $62.4 \pm 1.2$ | $63.7 \pm 0.8$ (cosine-2) |
| Proteins | $70.9 \pm 0.4$ | $72.7 \pm 0.5$ | $71.4 \pm 0.7$ | $73.0 \pm 0.6$ (cosine-rlb-1) |
| NCI1 | $74.1 \pm 0.3$ | $73.3 \pm 0.3$ | $81.4 \pm 0.3$ | $76.4 \pm 0.3$ (cosine-rlb-1) |
| MSRC | $92.9 \pm 0.8$ | $91.2 \pm 0.9$ | $91.0 \pm 0.7$ | $95.2 \pm 1.3$ (poly-1) |
| BZR | $81.9 \pm 0.6$ | $81.4 \pm 1.2$ | $85.9 \pm 0.9$ | $85.1 \pm 1.1$ (poly-rlb-2) |
| COX2 | $78.4 \pm 1.0$ | $79.6 \pm 1.1$ | $80.7 \pm 0.8$ | $81.8 \pm 2.1$ (poly-rlb-1) |
| DHFR | $79.1 \pm 1.0$ | $79.2 \pm 0.7$ | $81.4 \pm 0.6$ | $80.1 \pm 0.5$ (poly-rlb-3) |

Table 1: Classification accuracies for general labeled graphs (the 1-gram case).

As discussed in the paper, we obtained best results for $p = 1$. However, we also sketched the explicit feature maps for the polynomial and cosine kernels for $p = 2$. (Note that the running time for KONG in Table 2 in the main body of the paper also include the time for sketching.) We present classification accuracy results for sketch sizes 100, 250, 500, 1000, 2500, 5000 and 10000 in Figure 1. As evident from the values, the values are close to the case $p = 1$ and also for quite small sketch sizes we obtain good accuracy. This indicates that sketching captures essential characteristics of the 2-gram frequency distribution also for small sketch sizes and can indeed yield compact feature maps.

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

Figure 1: Comparison of classification accuracy for graphs with ordered neighborhoods.