[Reviews · NeurIPS 2018]

Reviewer 1



The paper proposes a family of kernels for graphs with ordered neighbourhoods and discrete labels. Any member of the family is obtained by generating a string-based representation of each node of a graph and a subsequent exploitation of the basic spectrum string kernel by Leslie et al. PSB 2002. Specifically, the string associated to a node of the graph is efficiently and recursively generated via a tree traversal method that uses the neighbour ordering. The visit starts from a node and ends when the depth of the tree is h. Thus the visited subtrees are the starting features exploited by the kernel. The string is obtained by concatenating the discrete information attached to visited nodes. Contrary to what happens for recent state-of-the-art graph kernels, the string may not be hashed, but a k-gram histogram of it is computed. Since the number of different k-grams can quickly grow as a function of node degree and alphabet size, it is proposed to use a sketch strategy to just keep the hitters. The adopted technique is Tensor-Sketch (by Pham & Pagh, KDD 2013), which allows to have acceptable computational times. Moreover, it is observed that the histograms at depth i can be updated incrementally from those at level i-1 plus information on the (i-1)-prefixes and (i-1)-suffixes. An explicit feature space representation for the graph is thus obtained by summing up all the histograms obtained for the graph nodes. It is also observed that: i) kernels such as cosine and polynomial can be applied to histograms to obtain a soft matching; ii) the proposed family of kernels, due to their incremental nature, can also the adopted for streams of graphs. Quality: The mathematical treatment is solid as well as all the concepts that are used as building blocks of the proposed approach. I just have few concerns on the following points: i) the tree traversal strategy shapes the basic subtree features used by the kernel. This is not discussed in the paper. For example, while it is clear what the subtrees are for Algorithm 3 (weisfeiler-lehman kernel), this is not the case for Algorithm 2. In fact, breadth-first search skips edges between nodes at the same depth. This issue is deeply discussed in the paper (not cited) proposing the ODD kernel (A Tree-Based Kernel for Graphs. SDM 2012: 975-986). In the same paper, the issue about the ordering is addressed. In fact, in order to use tree kernels, it is important to have ordered children. The effort, in that case, was to give an order in a principled way, so to avoid different representations for isomorphic graphs. Clearly, if an order is given, than ODD kernels can be directly used to deal with ordered neighbourhoods. This is also the case for the WL kernel, where instead of using the lexicographic order to sort the labels of the neighbourhoods, the given order can be used. This trivial adaptation of existing kernels is not discussed in the paper. A related issue is how the experimental results for classical kernels reported in table 2 have been obtained: is the given order exploited by them or not ? The reverse question is valid for the results reported in Table 1. ii) it is not clear to me how much of the experimental results are due to the use of cosine and polynomial kernels. To make a fair experimental comparison, also the other kernels should be composed with cosine and polynomial kernels. Moreover, I found the range of values for the WL kernel hyper-parameter too narrow, since my experience with that kernel on exactly the same datasets of the first part of the experiments is that optimal values, for some of them, are outside that range, although I use a nested CV approach for hyper-parameter selection. iii) A relevant graph kernel for graphs exploiting a sketching approach (lossy-counting) has been proposed in (A Lossy Counting Based Approach for Learning on Streams of Graphs on a Budget. IJCAI 2013: 1294-1301). It would be nice to elaborate on the differences with respect to the proposed approach. Minor issues: i) it would be useful to report variance for all curves plotted in figure 2. ii) the column referring to computational times in table 2 is confusing: it is not readily clear why there are two numbers for each entry; the heading of the table, as well as the associated caption, should explain that. iii) typos: 109: “for same neighborhood”; line 207: “representation for the explicit feature maps”; line 245: “obtain following result”; line 246: “be stream”; 293: “the the”. Clarity: the paper is generally well written and can be followed with no problem by an expert in the field. Organisation of the material is good. The section I think would need better explanations is the one on experiments. In particular, the issues I have raised at the end of point i) above should be clarified. Originality: the paper builds on results from the literature. The idea to just use the k-gram histograms of the subtree visits is moderately new to my knowledge, although it can be considered a bit derivative. Moreover, I am not convinced that this strategy can work well for larger values of h unless large values of k are used. Some effort has been put into the derivation of the theoretical results. Overall I see some original contribution although limited. In particular, I think the paper still misses some deeper analysis of the generated feature spaces. Significance: I guess the main contribution of the proposed kernels is given by the fact that soft matching is introduced both at the level of histograms and explicit feature vector approximation. Possible follow-ups seem to be limited to the ones already described in the paper. After rebuttal. Concerning fair experimental comparison: most of the more advanced graph kernels nowadays use explicit representations. This allows a simple application of a polynomial kernel, as envisaged in the last part of the answer. I still think a fair comparison should compare against this version of the base kernel. This can be easily done for WL.

Reviewer 2



The paper presents a new graph kernel for the setting where the edges incident to a node have associated order. The paper combines the WL-type approaches with sketching of node feature vectors, and provides an efficient update algorithm for label k-gram distributions of the nodes through dynamic programming. The authors further provide theoretical analysis of the complexity of the sketching approach. The experimental results are convincing especially for the graphs with the ordered neighbourhood. Update: I have read the author response

Reviewer 3



The manuscript proposes a novel graph kernel for labelled graphs with ordered neighborhood. The main idea of the proposed (KONG) framework is to explicitely represent graphs using a particular feature map. This mapping is done by representing each node with a string representation derived from its neighborhood using a sketching technique to get compact represenations. Finally, the graph is the overall sum of the representation of its nodes. Authors provides also proofs about the complexity of the proposed algorithms and some bounds about the approximation made by using sketching methods. Finally, experiments with both standard graphs and neighborhood ordered graphs are desribed, showing the efficiency and effectiveness of the proposal. The paper is clear and well written. The structure is ok and provides all the necessary background to ease the reader throughout the paper. In my opinion, the related work part is a bit weak and I would suggest to add some other references, e.g., - Graph Kernels Exploiting Weisfeiler-Lehman Graph Isomorphism Test Extensions, Giovanni Da San Martino, Nicolò Navarin and Alessandro Sperduti. In Neural Information Processing, Lecture Notes in Computer Science, Volume 8835, 2014, pp 93-100. - A Lossy Counting Based Approach for Learning on Streams of Graphs on a Budget, Giovanni Da San Martino, Nicolò Navarin, Alessandro Sperduti. In 23rd. International Joint Conference on Artificial Intelligence, August 3-9, 2013 - Beijing, China. The technical part seems sound to me, and I've appreciated that all the proofs are in a separate Appendix. Regarding the experimental section I have the following comments: - It is not clear why the authors have selected those baselines. Which was the rationale behind such choice? - I've noticed that between the experiments with general graphs and ordered neighborhood graphs the validated ranges for the SVM hyper-parameter C are different. Why? In particular, in the case of general graph the range seems quite small to me; - I would suggest to add a table with some information (#of classes, #of graphs...) about the used datasets even though they are pretty standard ones; - In Table 1, authors provide, inside the parenthesis, the best performing parameters. It is not clear to me whether the parameters have been selected a-posteriori or they have been validated. In the latter case should not be possible to give best performing parameters (each validation has its own best parameters); - Table 2: Since the experiments have been repeated several times the standard deviation should be reported (as in the previous case); - Table 2: It is not clear why KL has no AUC and accuracy for Movielens. = = = = = = = = = = Thank you to the authors for the response. I think this paper is a good paper and I confirm my evaluation.